# Efficacy of the Vaccine Candidate Based on the P0 Peptide against *Dermacentor nitens* and *Ixodes ricinus* Ticks

**DOI:** 10.3390/pathogens12111365

**Published:** 2023-11-17

**Authors:** Alina Rodríguez-Mallon, Pedro E. Encinosa Guzmán, Yamil Bello, Ana Domingos, Sandra Antunes, Petr Kopacek, Ana Sofia Santos, Rita Velez, Jan Perner, Frank L. Ledesma Bravo, Helena Frantova, Jan Erhart, Rafmary Rodríguez, Alier Fuentes, David Diago, Marisdania Joglar, Luis Méndez, Mario Pablo Estrada

**Affiliations:** 1Center for Genetic Engineering and Biotechnology (CIGB), 31st Avenue and 190, Havana 10600, Cuba; pedroencinosa88@gmail.com (P.E.E.G.); frank.ledesma@cigb.edu.cu (F.L.L.B.); david.diago@cigb.edu.cu (D.D.); marisdania.joglar@cigb.edu.cu (M.J.); mario.pablo@cigb.edu.cu (M.P.E.); 2Global Health and Tropical Medicine, GHTM, Associate Laboratory in Translation and Innovation towards Global Health, LA-REAL, Instituto de Higiene e Medicina Tropical, IHMT, Universidade Nova de Lisboa, UNL, Rua da Junqueira 100, 1349-008 Lisboa, Portugal; adomingos@ihmt.unl.pt (A.D.); santunes@ihmt.unl.pt (S.A.); 3Institute of Parasitology, Biology Centre, Czech Academy of Sciences, 37005 České Budějovice, Czech Republic; kopajz@paru.cas.cz (P.K.); perner@paru.cas.cz (J.P.); frantova@paru.cas.cz (H.F.);; 4Centro de Estudos de Vetores e Doenças Infeciosas Dr. Francisco Cambournac, Instituto Nacional de Saúde Doutor Ricardo Jorge (CEVDI-INSA), 2965-575 Águas de Moura, Portugal; ana.santos@insa.min-saude.pt (A.S.S.); rita.perdigao.velez@gmail.com (R.V.); 5Instituto de Saúde Ambiental, Faculdade de Medicina, Universidade de Lisboa, 1649-004 Lisboa, Portugal; 6National Laboratory of Parasitology, Avenue San Antonio-Rincón, Km 1 1/2, Artemisa 32500, Cuba; rafmaryr123@gmail.com (R.R.); alierfuentesc1315@gmail.com (A.F.);

**Keywords:** ticks, P0 protein, anti-tick vaccine, tick control, vaccination

## Abstract

The control of ticks through vaccination offers a sustainable alternative to the use of chemicals that cause contamination and the selection of resistant tick strains. However, only a limited number of anti-tick vaccines have reached commercial realization. In this sense, an antigen effective against different tick species is a desirable target for developing such vaccines. A peptide derived from the tick P0 protein (pP0) conjugated to a carrier protein has been demonstrated to be effective against the *Rhipicephalus microplus, Rhipicephalus sanguineus*, and *Amblyomma mixtum* tick species. The aim of this work was to assess the efficacy of this peptide when conjugated to the Bm86 protein against *Dermacentor nitens* and *Ixodes ricinus* ticks. An RNAi experiment using P0 dsRNA from *I. ricinus* showed a dramatic reduction in the feeding of injected female ticks on guinea pigs. In the follow-up vaccination experiments, rabbits were immunized with the pP0-Bm86 conjugate and challenged simultaneously with larvae, nymphs, and the adults of *I. ricinus* ticks. In the same way, horses were immunized with the pP0-Bm86 conjugate and challenged with *D. nitens* larva. The pP0-Bm86 conjugate showed efficacies of 63% and 55% against *I. ricinus* and *D. nitens* ticks, respectively. These results, combined with previous reports of efficacy for this conjugate, show the promising potential for its development as a broad-spectrum anti-tick vaccine.

## 1. Introduction

Chemicals used for tick control cause food contamination and environmental pollution [1,2,3,4]. In addition to these drawbacks, there is an increase in reports of multi-resistant tick strains to chemical acaricides [5,6,7,8]. In this current situation, vaccination becomes a very attractive alternative to control these ectoparasites. However, the identification of new effective antigens against several tick species and the technological development for obtaining them in a cost-effective manner appears very relevant in order to improve the practical application of anti-tick vaccines. In fact, only vaccines based on the Bm86 antigen have reached the market and have been applied against *R. microplus* ticks in the field [9,10,11,12].

In Cuba, 34 species of ticks have been described [13]. Of these, nine species belong to the Ixodidae family, and only four are considered important from economic and public health perspectives. One of them is the *Rhipicephalus microplus* (Canestrini, 1888), which constitutes the main vector of hemoparasitic diseases for bovines in the country, causing severe damage to livestock production [14]. *Rhipicephalus linnaei* (Audouin, 1826) [15], previously known as the tropical lineage of *Rhiphicephalus sanguineus*, is also present on the island [16,17]. It is one of the most studied tick species in the world due to its public health implications. Besides dogs, they can be found on a diverse range of wild and domestic animals, including humans [18,19,20], and some pathogens that they transmit are zoonotic [20]. *Amblyomma mixtum* (Koch, 1844) is another important tick species found in Cuba [21]. They inhabit peridomestic environments and have a three-host life cycle, parasitizing humans, livestock, and wildlife. They are also known vectors for important pathogens transmitted to humans and animals [22]. Finally, *Dermacentor nitens* (Neuman, 1897) is the main vector of piroplasmosis for livestock in Cuba, primarily including cattle and equids [22].

On the other hand, *Ixodes ricinus* (Linnaeus, 1758) species is found across Europe, North Africa, and the Middle East. It is a vector for different pathogens causing severe diseases in a variety of mammalian hosts [23]. Among the most important diseases transmitted by this tick species are viral encephalitis and zoonotic Lyme borreliosis, and it is also possibly responsible for alpha-gal allergy [24].

The objective of this study is to investigate the efficacy of the 20 amino acid peptide from the P0 acidic ribosomal protein of ticks (pP0) against *D. nitens* in immunized horses and against *I. ricinus* in immunized rabbits. This peptide had been selected from the P0 protein sequence of ticks as a vaccine antigen due to its lowest percentage of sequence identity with the orthologous protein of the mammalian hosts to avoid cross-recognition [25]. The use of the whole P0 protein as a vaccine antigen could be limited because it is a very conserved protein with an important role as the backbone of the 60S ribosomal subunit [26]. In this study, in order to improve the P0 peptide immunogenicity, it was conjugated to the Bm86 protein produced by *Komagataella (Pichia) pastoris* yeast [27] as a carrier protein. This protein is the antigen of the Gavac^®^ vaccine (Havana, Cuba) that has been proven to induce a strong immune response in vaccinated cattle and is effective against *R. microplus* ticks in field applications [9,28]. The use of this Bm86 protein produced by *K. pastoris* yeast in the conditions previously described has been suggested as an adjuvant for the formulation of combined vaccines in cattle, considering the high grade of aggregation in which it is obtained [29]. In previous studies, this peptide, when conjugated with other different carrier proteins, demonstrated efficacy against *R. microplus*, *R. linnaei*, and *A. mixtum* ticks [25,30,31,32,33]. All these previous results combined with those from this study allow us to suggest if this antigen could be effectively used in field conditions to control different tick species via host immunization.

## 2. Results

### 2.1. P0 Knockdown Impairs Feeding of I. ricinus Ticks

The injection of *I. ricinus* female ticks with IrP0 dsRNA caused the sequence-specific degradation of the P0 mRNA, efficiently suppressing the expression of this gene. This was demonstrated by the results of the quantitative real-time PCR performed on cDNA obtained from samples of the gut, salivary glands, and fat bodies of injected ticks (Figure 1). This RNAi-mediated silencing of IrP0 expression was manifested by the loss of some of the tick’s biological functions, which is in agreement with the fact that the P0 protein is a part of the biosynthetic machinery in the cells and could play an important role in these tissues during the initial feeding phase. The IrP0-silenced ticks had a profoundly decreased blood-feeding capacity, which resulted in a statistically significant diminution of engorged female weights compared to those of the control ticks after ten days of feeding (Figure 2). The gain weight of the IrP0-silenced female ticks represented about 3% of the weight of fully engorged female ticks in the control group injected with non-related GFP dsRNA. No oviposition or a very limited quantity of eggs that did not hatch were obtained from female ticks with silenced P0 expression.

### 2.2. The Vaccination of Rabbits and Horses with the pP0-Bm86 Conjugate Generates a Specific Antibody Response

After the immunizations of rabbits and horses with the pP0-Bm86 conjugate, the animals developed an immune response characterized by antibody titers of a specific total IgGs against vaccine antigens above serum dilutions of 1:1000 on day 36, which lasted until the end of the experiments (Figure 3). By day 21 after the initial immunization, all vaccinated animals, except one horse, had developed a specific antibody response that reached its peak level on day 36 in rabbits and on day 51 in horses. On the day of the tick challenge (Day 51) for both experiments, the animals had antibody titers above serum dilutions of 1:1000 for both vaccine antigens.

### 2.3. The pP0-Bm86 Conjugate Used as Vaccine Antigen Is Effective against I. ricinus Ticks

The average number of fed *I. ricinus* larvae obtained from rabbits immunized with the pP0-Bm86 conjugate and their mortality during molting to nymphs did not show any statistically significant differences compared with these parameters in the control group (Table 1). However, standard deviations (SD) for both parameters in the immunized group were significantly different from those in the control group. In this case, where these parameters showed considerable variability inside the immunized group, it might be necessary to use a larger sample size than three in order to obtain statistical significance between these sample means.

On the other hand, the recovery average of fed nymphs from immunized rabbits and their mortality during molting to adults showed statistically significant differences compared with those parameters in the control group (Table 1). In the case of adults, non-statistically significant differences were found for the recovery average of engorged females and in the weights of egg masses. The hatching of eggs could not be determined because of an incubator breakdown that produced egg dehydration, unfortunately leading to the loss of this parameter. Despite this inconvenience, the overall efficacy of the pP0-Bm86 conjugate against *I. ricinus* ticks calculated only considering the recorded parameters that displayed statistically significant differences between the experimental groups, was of 63%, as shown in Table 1.

### 2.4. The pP0-Bm86 Conjugate Used as Vaccine Antigen Is Effective against D. nitens Ticks

The average number of fully engorged female ticks of *D. nitens* collected from horses immunized with the pP0-Bm86 conjugate did not show statistically significant differences when compared to the control group despite the fact that average number of engorged females obtained from the vaccinated group was only around half of those obtained in the control group (Table 2). The high variability shown by this parameter within both groups, along with the small sample size (n = 5), could potentially make it harder to detect real differences through statistical testing. Interestingly, the weight of engorged females in the vaccinated group was statistically higher than in the control group; however, the egg masses laid by females in both groups were similar. It means that there was a significant difference in the conversion efficiency index, defined as the proportion of female weight that is converted to egg masses, as shown in Table 2. The hatching of these egg masses in the vaccinated group demonstrated a highly statistically significant reduction compared to the control group. The calculated overall efficacy of the pP0-Bm86 conjugate against *D. nitens* ticks was 55%, considering only parameters showing significant differences between the experimental groups. As *D. nitens* is a one-host tick, the effects of the pP0-Bm86 vaccination on larva and nymph stages should be included in the effects observed on female ticks.

## 3. Discussion

The dramatic effect of the P0 knockdown on *I. ricinus* feeding that was obtained by the RNAi experiment described in this paper, corroborates the previous results obtained for other tick species. These studies demonstrated that the P0 ribosomal protein is essential for blood ingestion and tick viability [34,35,36]. P0 is a multifunctional protein whose best-known biological function is its protagonist structural role in the 60S ribosomal subunit assembly in all living cells [37]. However, its endonuclease activity in cellular nucleus when dephosphorylated has also been described [26,38], as well as its localization at the surface of various eukaryotic cells, including parasites [39,40]; more recently, it was found in tick saliva [41,42]. Logically, the observed phenotype in P0 knockdown ticks is a result of the disruption of all these known biological functions but also those potentially unknown functions in the cellular membranes and in tick saliva. Nevertheless, these effects mediated by RNAi may not necessarily be the same when the P0 protein is used as a vaccine antigen. The efficacy of an anti-tick vaccine depends on whether specific antibodies and other immune response mediators induced by host vaccination are capable of reaching the target protein in its cellular locations and which of its biological functions are blocked by that specific immune response. In fact, the anti-tick effect observed when the pP0 is used as a vaccine antigen is not as dramatic on feeding and viability as when IrP0 dsRNA is injected in ticks, as shown in Figure 2.

Previous studies using the pP0 as an anti-tick vaccine antigen have reported around a 90% efficacy against *Rhipicephalus* ticks [25,30,32] and more than a 50% efficacy against *A. mixtum* larvae [31], mainly affecting the molting processes, oviposition, and egg hatchability. In this work, the main effect of the pP0-Bm86 vaccination on *I. ricinus* ticks was in the nymph stage and in molting adults. Unfortunately, we missed the opportunity to evaluate egg hatching in this experiment. Despite this inconvenience, the calculated efficacy for the pP0-Bm86 conjugate against *I. ricinus* ticks was 63%. On the other hand, the main outcome of feeding *D. nitens* ticks on vaccinated horses with the pP0-Bm86 conjugate was a significant reduction in egg hatchability with an overall calculated efficacy of 55%. Further experiments need to be conducted to elucidate which biological functions of the P0 protein can be blocked by the specific immune responses induced in the vaccinated hosts. We have some experimental evidence of preliminary proteomic studies from the organs of these *D. nitens* ticks that need to be corroborated, indicating that the protein synthesis is not the main target of the P0 function blocked by these specific immune responses against pP0 (unpublished results).

Host vaccination with the Bm86 antigen has been demonstrated to be effective against *R. microplus* ticks; however, rabbit vaccination with two Bm86 homologs in *I. ricinus* (Ir86-1 and Ir86-2) did not show efficacy in control populations of this tick species [43]. Other previous studies have suggested that the variability in the Bm86 protein sequence could be one of the major factors that influences the effectiveness of this antigen against different tick species [44,45]. Phylogenetic studies demonstrated that the Bm86 coding sequence from *R. microplus* is not closely related to the sequences from species of the *Amblyomma, Dermacentor*, *Haemaphysalis*, and *Ixodes* genus [45]. For these reasons, even though high antibody titers were obtained in vaccinated animals against the Bm86 protein, as expected due to its very good previously demonstrated immunogenic properties [46], the efficacy against *I. ricinus* and *D. nitens* ticks obtained in this study could be attributed to antibodies against pP0. In addition, our previous results also demonstrated that the conjugation of the pP0 to the Bm86 produced by yeast as a carrier protein is able to enhance the immune response against pP0; however, under the specific conditions of conjugation in which the N-β-(maleimidopropyloxy) succinimide ester (BMPS) was used as a heterobifunctional cross-linker between the primary amino groups of the carrier protein and the thiol group of the Cys residue intentionally added at the N-terminal end of the pP0, there was no increase in the efficacy obtained against ticks compared to those obtained when the pP0 was conjugated to other carrier proteins [32]. This result contradicts what could be expected of a vaccine candidate in which two antigens against ticks are combined. It is suggested that the conjugation could eliminate at least to some extent, important epitopes that are responsible for the protective effect of the Bm86 antigen against ticks.

The results shown here, in conjunction with previous efficacy reports for a vaccine candidate based on the P0 peptide against ticks, validate our initial hypothesis that this could be an anti-tick antigen with a broad spectrum of action. This hypothesis took into account the high sequence identity (more than 80%) of this peptide among different tick species [30]. In addition to the present results and before undertaking production and commercialization, further dosage studies for different hosts and evaluations under field conditions are necessary. Our experience with the Bm86 antigen showed that protection with the Gavac^TM^ (Havana, Cuba) vaccine is higher in the field than that observed in controlled trials [47]. If this were the case with the pP0-Bm86 antigen, it could be used to control the four most important tick species in Cuba from both economic and public health perspectives, as well as *I. ricinus*, which is an abundant tick species in Europe and an important vector for zoonotic pathogens. Further experiments should also be performed in order to evaluate if immunization with the P0-Bm86 conjugate could affect pathogen transmission by ticks.

## 4. Materials and Methods

### 4.1. Ticks and Animals

Female and male ticks of *I. ricinus* collected in the forest around Ceske Budejovice (48°54′32″ N13°53′12″ E), Czech Republic, were used for the RNAi experiment on guinea pigs. The ticks were maintained in wet chambers with a humidity of about 95%, at a temperature of 24 °C, and a day/night cycle set to 15/9 h. All laboratory animals were handled in accordance with the Animal Protection Law of the Czech Republic No. 246/1992 Sb., with ethics approval No. 25/2020 issued by the Institute of Parasitology.

Larvae, nymphs, and adults of *I. ricinus* ticks used in the experiment of immunization and the challenge of rabbits were obtained from a colony established on behalf of the FCT project—TickGenoMi PTDC/SAU-PAR/28947/2017, at the Center for Vectors and Infectious Diseases Research, National Institute of Health Dr. Ricardo Jorge (CEVDI-INSA) in Portugal. Two groups of three Hyla rabbits each, of both sexes, weighing around 2.5 kg, were provided with a commercial diet and water ad libitum during the experiment. All procedures involving animals followed the Guide for the Care and Use of Laboratory Animals [48] and were approved by the Ethics Committee of the Institute of Hygiene and Tropical Medicine of the Lisbon University in Portugal, where this experiment was performed. The rabbit experiment was conducted with the approval of the Divisão Geral de Alimentação e Veterinária (DGAV), Portugal, under Art◦ 49, Portaria n◦1005/92 from August the 7th (permit number 0421/2022).

Larvae of *Dermacentor nitens* ticks used in the immunization and challenge experiments of horses were obtained from a colony established at the Cuban National Laboratory of Parasitology [49]. This colony was established from an engorged female tick obtained from the perianal region of a horse in the Bauta municipality in Cuba (22°59′31″ N, 82°32′57″ W). Young Quarter horses were obtained from a livestock company in the San Cristobal municipality in the Pinar del Río Province in Cuba (22°42′59.69″ N, 83°03′23.29″ W). They were individually stabled during the experiment performed in the company facilities and were fed with a dairy diet corresponding to 2% of their body weight in forage, supplemented with 2 kg of concentrate feed twice in the day and water ad libitum. The sampling exercise and all procedures were performed under the Cuban Decree-Law 31/2021 “On Animal Welfare” (GOC-2021-332-EX25) and were approved by the Ethics Committee at the CIGB (#22040).

### 4.2. RNA Interference

A 326 bp DNA fragment was PCR-amplified using cDNA from fat bodies of semi-engorged *I. ricinus* female ticks as a template and specific primers were designed from the reported P0 sequence of *I. ricinus* in Genbank (Accession number: GADI01004298.1) (Forward: 5′...ATGGGCCCTGCAGATCCCCACCAAGATC...3′ and Reverse: 5′...ATTCTAGACCGTTCACGATGGAGTGTG...3′) containing Apa I and Xba I restriction sites and using Primer3 software (version 4) [50]. The specific restriction sites were used to clone the amplified DNA fragment. The amplicons were purified from 2% agarose gel in a TA1X buffer using a Gel and PCR Clean-up Kit (Macherey-Nagel). The IrP0 fragment and the green fluorescent protein (GFP) dsRNAs were synthesized using the MEGAscript T7 transcription kit (Ambion; Vilnius, Lithuania) according to the previously described protocol [51]. Unfed *I. ricinus* female ticks were injected with 0.5 μL (3 μg/μL) of IrP0 dsRNA or GFP dsRNA in the control group and were maintained at 24 °C for 24 h. Afterward, 30 female ticks from each group were allowed to feed naturally on guinea pigs in the presence of an equal number of males until repletion. After two post-infestation days, three ticks of each group (GFP and IrP0-dsRNA injected) were randomly collected to examine the efficiency of P0 RNAi knockdown via reverse transcription and quantitative real-time PCR (qRT-PCR) using the following primers: F: CCAAGAAGGAGGAGGCTAAGA; R: TAGTCAAAGAGGCCGAATCC in midgut, salivary gland and fat body samples. The P0 transcription levels in all samples were determined relative to the expression of the elongation factor 1 alpha (EF1A, GenBank ID: GU074769) as a reference gene. The other injected *I. ricinus* female ticks fed on guinea pigs were observed daily until day 10, in which engorged female ticks were collected and weighed. A weight comparison was performed by an unpaired Student t-test using the GraphPad Prism program (version 9.4 for Windows, Royal, AR, USA).

### 4.3. Synthesis of pP0-Bm86 Conjugate

The P0 peptide (NH_2_-AAGGGAAAAKPEESKKEEAK-CONH_2_, pP0) obtained by chemical synthesis with a Cys residue added to its N terminal end (CS Bio Company, Shanghai, China, Lot: CS8O15042401) was conjugated to the Bm86 protein produced by *Komagataella (Pichia) pastoris* yeast (Gavac^TM^ active ingredient, Havana, Cuba Lot: 14.1903-12) by the BMPS method as previously described [32].

### 4.4. Immunization and Challenge Experiment in Rabbits

Three rabbits received three subcutaneous immunizations on days 0, 21, and 36 with 1 mL of an oily formulation containing 500 μg of the pP0-Bm86 conjugate adjuvanted in Montanide ISA50 (SEPPIC, France) in a 60/40 proportion of aqueous /oily phases (*w*/*w*). The other three rabbits received, on the same days, injections of PBS in the same oily preparation as the vaccine antigen. Fifteen days after the last immunization, all rabbits were challenged with 300 larvae, 25 nymphs, and 35 adults (20 females and 15 males) of *Ixodes ricinus* ticks in an independent craft chamber for each life stage. All fed stages were collected as they spontaneously detached from the animal. Larvae and nymphs were counted, weighed, and incubated at 23 °C and 95% relative humidity until molting to the next instar. The adults were counted, weighed, and incubated at 23 °C and 95% relative humidity for 20 days during oviposition in which the egg masses were weighed. Eggs were incubated in the same conditions until hatching. The hatching percentage was estimated using a visual evaluation as previously described [52]. The overall efficacy of the pP0-Bm86 conjugate was calculated according to the following formula:E = 100 × (1 − [RL × VL × RN × VN × RA × PA × FE])
where RL and VL are the parameters related to the recovery of fed larvae and mortality in the molting process of the vaccinated group compared to the control; RN and VN are the parameters related to the recovery of fed nymphs and mortality in the molt of the vaccinated group compared to the control. RA and PA are the effects of the immunogen on female recovery and oviposition (weight of egg masses) compared to the control group. FE is the effect of immunogen on egg fertility. It is calculated as the ratio between the hatching percentages of eggs laid by ticks fed on vaccinated animals compared to the hatching percentage in the control group. All average comparisons between data from the experimental groups were performed using an unpaired Student t-test and the GraphPad Prism program (version 9.4 for Windows, Royal, AR, USA). Parameters in vaccinated groups that did not show statistically significant differences compared to those in the control group were considered equals in their efficacy calculations.

### 4.5. Immunization and Challenge Experiment in Horses

Ten horses were randomly assigned to two experimental groups, with five animals in each one. One group was subcutaneously immunized with 2 mL of a vaccine formulation prepared with 250 μg/mL of the pP0-Bm86 conjugate adjuvanted in 5% of Montanide™ GEL 01 (SEPPIC, Puteaux, France) on days 0, 21 and 36. The other group was injected with 2 mL of PBS in the same formulation in a schedule that immunized horses. Fifteen days after the last immunization, all horses were challenged with *D. nitens* larvae from 1 g of the eggs (around 3000 larvae). Fully engorged female ticks were collected as they spontaneously detached from the animal around 18 days after larva infestation. They were weighed in groups of 20, as were the egg masses. The hatching percentage was estimated by a visual double-blind evaluation. The pP0-Bm86 conjugate efficacy was calculated according to the following formula:E = 100 × (1 − [RA × PA × FE]) 
where RA and PA are the effects of immunogen on female recovery and oviposition (weight of egg masses) compared to the control group, respectively. FE is the effect of each immunogen on egg fertility. It was calculated as the ratio between the hatching percentage of eggs laid by ticks that fed on vaccinated animals compared to the hatching percentage in the control group. Average comparisons among groups were performed using an unpaired Student t-test and using the GraphPad Prism program (version 9.4 for Windows, USA).

### 4.6. Antibody Response Evaluation

Serum samples from rabbits and horses were taken on days 0, 21, 36, and 51 in order to measure antibody responses against vaccine antigens via an indirect ELISA. In the case of rabbits, serum samples on day 61 were also taken. Bm86 and pP0 conjugated to a non-related carrier protein were indistinctly used to coat ELISA plates overnight at 4 °C in a coating buffer (Na_2_CO_3_ 1.5 g/L; NaHCO_3_ 2.93 g/L, pH 9.6). Our previous experiments evidenced that the pP0 sticks poorly to the ELISA plate. For this reason, the protocol of this indirect ELISA was established using the peptide conjugated to a carrier protein different from that used for immunization. In this way, specific antibodies against the P0 peptide could be detected. Clearly, specific antibodies against this non-related carrier protein should not be present in the animal sera when immunized with another conjugate. If some unspecific reactivity could be obtained, it would be blanked with the OD of the pre-immune animal sera. Afterward, plates were washed three times with Tween 20 at 0.05% in PBS and incubated for 2 h to 37 °C with a blocking solution (Skim milk at 5% in PBS). After washes, sera were added to plates serially twofold-diluted with PBS, beginning with a serum dilution of 1:100 and 1:500 for plates coated with the pP0 conjugate or Bm86, respectively. Plates were incubated for 1 h at 37 °C. Secondary antibodies were used anti-rabbit or anti-horse IgG–HRP conjugates (Sigma, Ronkonkoma, NY, USA) as appropriate, diluted at 1: 10,000 and incubated for 1 h at 37 °C. The color reaction was developed with a substrate solution containing 0.4 mg/mL of o-phenylenediamine in 0.1 M citric acid and 0.2 M Na_2_HPO_4_, pH 5.0, and 0.015% hydrogen peroxide. The reaction was stopped with 2.5 M H_2_SO_4_, and the OD 490 nm was determined. The antibody titer was established as the reciprocal of the highest dilution, at which the OD of the studied serum was three times the mean OD of the negative control serum. Titer results were presented as the geometric mean of each group.

## 5. Conclusions

The results revealed by this study, together with previous reports of efficacy shown by the P0 peptide conjugated to the Bm86 as a carrier protein, support the fact that this antigen could be a good candidate to develop an anti-tick vaccine to control different tick species.

## Figures and Tables

**Figure 1 pathogens-12-01365-f001:**
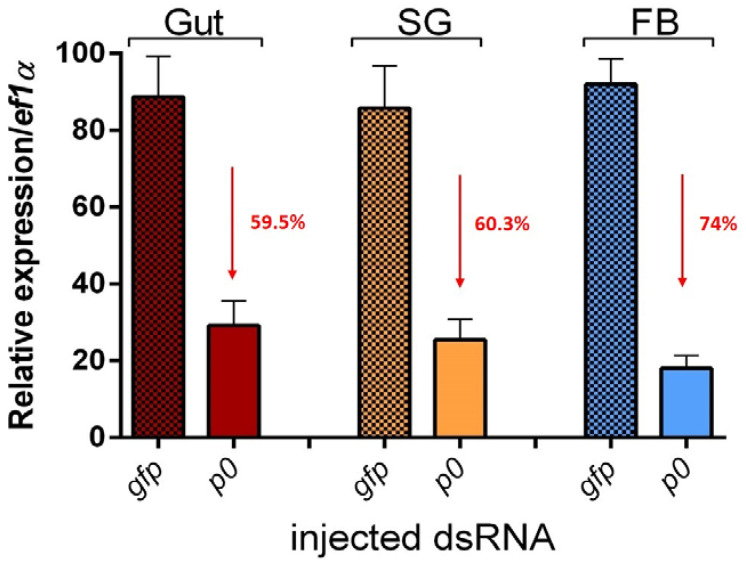
Quantitative real-time PCR assays showing the knockdown expression of the P0 gene in the gut, salivary glands, and fat body of *I. ricinus* female ticks injected with IrP0 dsRNA, relative to female ticks injected with GFP dsRNA as a non-related control. These P0 expression levels were normalized to the elongation factor 1 alpha (EF1A) mRNA. Error bars represent the standard deviation of the mean of 3 independent biological replicates. Percentages of reduction in the P0 gene expression are highlighted in red.

**Figure 2 pathogens-12-01365-f002:**
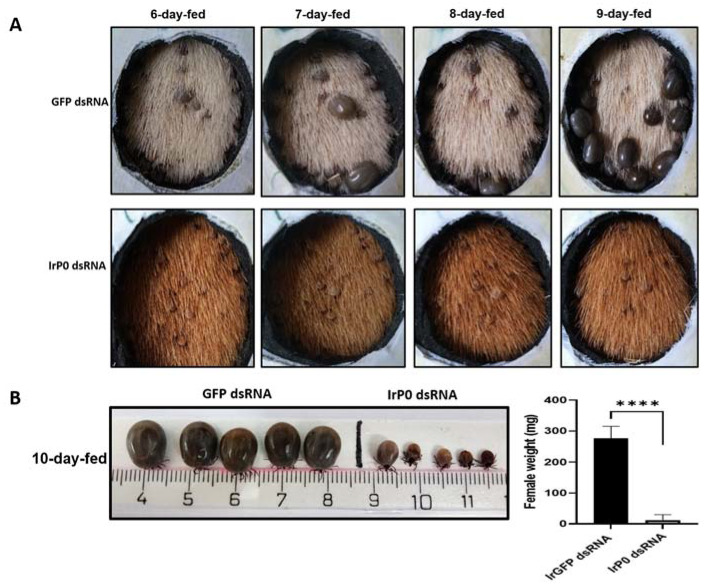
Engorgement parameters of *I. ricinus* female ticks injected with IrP0 dsRNA compared to female ticks injected with GFP dsRNA. (**A**) *I. ricinus* female ticks during feeding on guinea pigs. (**B**) Engorgement parameters of *I. ricinus* female ticks after 10 days of feeding. Asterisks represent highly significant statistical differences determined using an unpaired Student *t*-test (**** *p* < 0.0001).

**Figure 3 pathogens-12-01365-f003:**
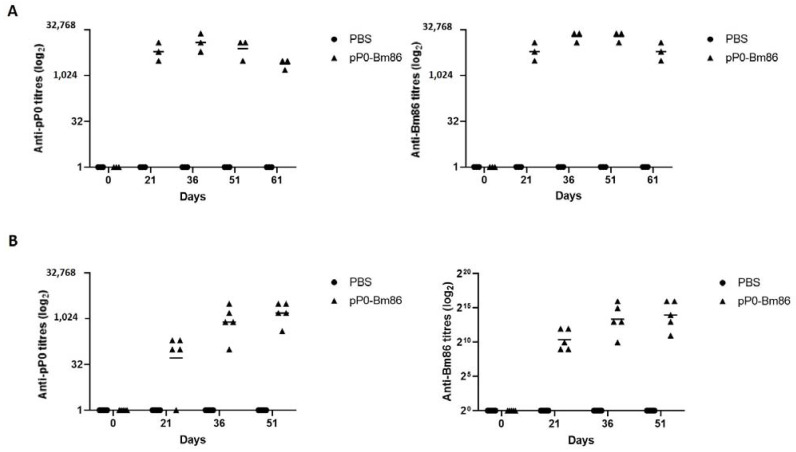
Specific response of total IgGs developed by animals immunized with the pP0-Bm86 conjugate on days 0, 21, and 36. These antibody titers were measured using an indirect ELISA. (**A**) Rabbits; (**B**) Horses. The antibody titer was established as the reciprocal of the highest dilution at which the OD of the studied serum by iELISA was three times the mean OD of the negative control serum. The lines in each experimental group represent the geometric mean of the antibody titers. Black circles and triangles represent the antibody titer of individual animals in the control and in the pP0-Bm86 immunized groups, respectively. The *Y* axes are in the logarithmic scale.

**Table 1 pathogens-12-01365-t001:** Effects of immunization with the pP0-Bm86 conjugate challenged with *I. ricinus* ticks in rabbits.

Group	Larva Yield	Mortality in Molt (%)	Nymph Yield	Mortality in Molt (%)	Female Yield	Female Weight (mg)	Egg Mass (mg)	Hatching (%)	E (%)
**pP0-Bm86**	142 ± 107 ^a^	65 ± 21 ^a^	16 ± 1 ^a^	86 ± 3 ^a^	17 ± 2 ^a^	207.5 ± 66.1 ^a^	40.57 ± 22.3 ^a^	ND	63
**PBS**	163 ± 50 ^a^	54 ± 5 ^a^	21 ± 2 ^b^	71 ± 8 ^b^	18 ± 1 ^a^	185.5 ± 55.3 ^a^	47.25 ± 20.3 ^a^	ND	

Average ± Standard deviation of each recorded parameter was included (n = 3 for each group). Different letters represent statistically significant differences determined by an unpaired Student *t* test (*p* < 0.05).

**Table 2 pathogens-12-01365-t002:** Effects of immunization with the pP0-Bm86 conjugate on challenges with *D. nitens* ticks in horses.

Group	Female Yield	Female Weight (mg)	Egg Mass (mg)	IEC (%)	Hatching (%)	E (%)
**pP0-Bm86**	373 ± 500 ^a^	302.2 ± 39.67 ^b^	142.3 ± 27.38 ^a^	47.42± 6.5 ^a^	68.67 ± 22.94 ^a^	**55**
**PBS**	693 ± 567 ^a^	276.1 ± 35.06 ^a^	142.5 ± 20.85 ^a^	49.97 ± 5.7 ^b^	82.71 ± 14.23 ^b^	

Average ± Standard deviation of each recorded parameter was included (n = 5 for each group). Different letters represent statistically significant differences determined by an unpaired Student *t* test (*p* < 0.05 for female weight and IEC) (*p* < 0.0001 for hatching).

## Data Availability

Data supporting reported results can be requested to authors.

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
