# Peer review of "Efficacy of the Vaccine Candidate Based on the P0 Peptide against Dermacentor nitens and Ixodes ricinus Ticks"

_pathogens, 2023, doi:10.3390/pathogens12111365_

Round 1
Reviewer 1 Report
Comments and Suggestions for Authors
In this paper authors have investigated roles of P0 tick peptide and effects of immunization with P0-derived peptide conjugated with tick midgut protein Bm86. This study is continuation of authors’ previous work. Although results being worthy, several issues have to be addressed (bullets below).
Title: …ON the P0 peptide…
· Line 17: Please elaborate this statement in one sentence in Introduction section; contamination of produced food, environment…
· Lines 17-18: This statement is nowhere else elaborated, please add this in Introduction section; also, actually how many anti-tick vaccines have reached the market?
· Lines 26, 27: larvae
· Lines 36-38: unclear sentence, please rephrase
· Line 62: produced as a protein
· Line 63: please add one sentence on Gavac vaccine (and its variants)
· Lines 59-68: Here as well should be written something about P0 tick protein, its roles, protective immunization. Which were other carrier proteins, were they immunogenic or not? (in only one paper, ref. 22 is mentioned „limpet haemocyanin“)? Why did authors choose Bm86 instead, to increase efficacy of immunization...?
· Fig. 2 – engorgement parameters rather than phenotype
· Lines 96, 101: maybe better format 1:1000; were there 500 dilutions in total checked?
· Line 103: what does recovered fed larvae mean? And at other places in the texts.
· Line 118: does this mean that RL and VL values were not included in calculation?
· Line 120: according to the equation in the “materials and methods” section – redundant
· Fig. 3 – how do the triangles represent pP0-Bm86 seroreactivity, if plates were coated with individual proteins?
· Table 1: please rephrase the title; missing a right part?
· Table 2: missing a right part
· Discussion should be a bit more elaborated. What is the contribution of Bm86 in immunoprotection (it cannot be attributed solely to P0)? Maybe conclusions should be toned-down a bit, because eventual immunization with P0-Bm86 would not affect feeding of ticks that much, i.e. pathogen transmission.
· Line 202: ON guinea pigs
· Line 206: issued by…
· Line 208: OF rabbits…?
· Line 210: how many groups, males/females?
· Line 218: Were larvae or females used for infestation of horses – Line 134? This is unclear throughout the text, for instance 292-294
· Lines 203, 245-246: why these organs were chosen?
· Line 231: which software/tool was used for primer designing?
· Line 232: oligonucleotide - redundant
· Line 234: What is the purpose of restriction sites in downstream protocol?
· Line 254: please describe briefly (what does “(CS Bio, Batch: 254
· CS8O15042401” mean?)
· Lines 206, 289: why is used same concentration of antigen for rabbits and horses (huge difference in body mass)?
· Line 270: what does it mean visual evaluation, please describe
· Line 287: male/females?
· Line 309: Why authors didn’t use sole Bm 86 and P0 proteins to coat plates? What is non-related carrier protein and was its immunogenicity analyzed? What is the coating buffer? What is a blocking solution? Why are sera diluted in PBS, not in PBSTween/TBSTween? 1:2 is a very low sera dilution.
Comments on the Quality of English LanguageLanguage editing is advised.
Author Response
Dear reviewer, thanks for your valuable contribution to this manuscript improvements.
The manuscript in its present form was modified according your suggestions in the following:
- The title was modified.
- A sentence was added to the Introduction section containing the statement about contamination produced by chemicals.
- A sentence was also added to the Introduction section addressing anti-tick vaccines that have reached the market and their respective references.
- All type mistakes pointed out by you were corrected.
- The sentence on lines 36-38 was rephrased.
- A sentence was also added to the Introduction section regarding Gavac vaccine and its corresponding references.
- Something about P0 tick protein was written in the introduction section. All references in which the P0 peptide conjugated a different carrier proteins was assayed were included (with KLH, p64K of Neisseria meningitidis and Bm86). In the paper published by Pathogens, 9(6), 513. DOI:10.3390/pathogens9060513 was explained why Bm86 was selected as the carrier protein to assume the technological development of the pP0 as anti-tick vaccine antigen. The most important reasons are related to the production cost because an important requirement for a veterinary vaccine is a low cost to be produced. More extensive explanations about the P0s roles and protective immunizations were included in the discussion section.
- The legend of Fig. 2 was modified according your suggestion.
- On Lines 96- 101 the antibody titers were expressed as suggested by you. In addition, a sentence in the ELISA description in the “Materials and Methods” section was included in which the starting serum dilution was specified in each case.
- The term “recovered” was changed throughout the text.
- The answer to your question: “Line 118: does this mean that RL and VL values were not included in calculation?” is yes. These parameters were included in the calculation, however as their averages did not give statistical significant differences each parameter in the equation was 1 by dividing RL in vaccinated/ RL in control and VL vaccinated/VL in control.
- The redundancy on Line 120 was eliminated.
- Your observation regarding that the ELISA plates were coated with individual proteins is correct. The legend of Fig. 3 declared that triangles represent seroreactivity in the group vaccinated with the pP0-Bm86 conjugate. However, the name in the axis y, specifies titers against the specific antigen.
- The table titles were rephrases and the missing right part of tables were also fixed.
- A paragraph was added to Discussion section in which the contribution of Bm86 in immunoprotection was discussed. A last sentence was also added to this section in which the need for further experiments in order to demonstrate if vaccination with this antigen not only affects tick vectors, but also pathogens transmitted by them.
- In the “Materials and Methods” section on Line 210, sex of rabbits was included. The quantity of rabbits in the experiment was described in the immunization protocol in the same section.
- The answer to your question: “Were larvae or females used for infestation of horses – Line 134? This is unclear throughout the text, for instance 292-294” is that Dermacentor nitens is a one host tick species as microplus. For this reason, we did the challenge with larvae and we collected fully engorged females. The effects on larva and nymph stages should be included in the effects on female ticks. The sentence on line 134 was rephrased in order to express more clearly that D. nitens larvae for the challenge were obtained from the colony established in the Cuban National Laboratory of Parasitology and that colony was established from a female tick of this species obtained from a horse in the field.
- The answer to your question: “Lines 203, 245-246: why these organs were chosen?” is that the experiment could be done using the whole ticks, however, these organs were selected in order to improve sensibility and also because in the feeding start, it is suggested that P0 as part of the biosynthetic machinery in the cells should play an important role in this tissues.
- The software used for primer designing was added.
- The word “oligonucleotide” on line 232 was eliminated.
- The purpose of restriction sites in downstream protocol on Line 234 was declared.
- On Line 254 the meaning of “CS Bio, Batch: 254CS8O15042401” was declared. It is the lot of synthetized peptide.
- Regarding your question about the same concentration of antigen used for rabbits and horses is true that there is a huge difference in body mass. It is clear that during the technological development of this vaccine candidate, dosage studies for target host species should be developed, however, in these concept demonstration experiments, we decided to use the high dose that has been assayed in rabbits, dogs and cattle independently of their body masses taking into account that peptides are bad immunogens.
- A reference was included in which the visual evaluation of hatching is described in detail.
- The answer to your question: “Line 309: Why authors didn’t use sole Bm86 and P0 proteins to coat plates?” is that we have a source for Bm86 protein because our institution produces it in yeast, however, we don’t have P0 protein, only a synthetic peptide of P0. In previous experiments we had evidences that the peptide sticks poorly to the ELISA plate. For this reason, the protocol of this indirect ELISA was established using the peptide conjugated to a carrier protein different to that used for immunization. In this way, you only will detect specific antibodies against the P0 peptide. Obviously, specific antibodies against this non-related carrier protein should not be present in the animal sera immunized with other conjugate. Anyway, if some unspecific reactivity could be obtained, it would be blanked with the OD of pre-immune animal sera. The composition of the coating buffer and blocking solution were added to the ELISA protocol.
Reviewer 2 Report
Comments and Suggestions for Authors
The paper by Rodriguez-Mallon et al. reports the 63% and 55% protective efficacy achieved against Ixodes ricinus and Dermacentor nitens, respectively, in rabbits and horses vaccinated with the pP0 peptide from Rhipicephalus ticks conjugated with the Bm86 protein (pP0-Bm86).
The introduction provides sufficient background information and a clear objective. The methods are adequate and well described, and the results are clearly exposed. These results enlarge the tick species and genus range that can be targeted with a pP0-based anti-tick vaccine by adding I. ricinus and D. nitens to R. sanguineus, R. microplus and Amblyomma mixtum. Thus, the results of this work are interesting.
Additionally, I would like add the following some points for further discussion.
First, the RNAi-based P0 gene knockout experiment carried out in I. ricinus adds value to the manuscript since it confirms this gene/protein is a very interesting target for immune interventions aimed at tick control also in I. ricinus, given the dramatic inhibition of female tick feeding achieved. Regarding this experiment:
1) Did the authors assess whether the eggs weight and hatching percentage (i.e, females oviposition and fertility) were also affected?. It would have been very interesting to check these processes too, but I haven’t find any data or comment regarding this point in the manuscript. If this was not assessed, why not?.
2) A similar experiment with D. nitens would have been also desirable. What do the authors think about this?
Second, vaccination with the pP0-Bm86 conjugate induced a very strong humoral immune response against the Bm86 antigen. How much could these antibodies have contributed to the protection observed? And why or why not?
Comments on the Quality of English LanguageSome editing of English language required to make some sentences more fluid and natural.
Author Response
Dear Reviewer,
Thanks for your valuable revision of this manuscript. It has contributed a lot to improve the manuscript. In its present form, the manuscript was modified in the following:
- According to your suggestion, a sentence in the results of RNAi experiment was added in which the no oviposition of female ticks injected with P0dsRNA was declared.
- Regarding a similar experiment of RNAi on nitens, it was not performed. However, as explained in the manuscript, the results of this experiment on I. ricinus ticks corroborated previous results on other tick species performed by other authors and were congruent with the important biological functions of P0 protein. For this reason, we did not consider to do the same experiment on D. nitens ticks.
- Regarding your question about the role of the very strong humoral immune response against the Bm86 antigen obtained and its contribution to the observed protection, it coincided with a question released by reviewer 1. A paragraph was added to the Discussion section in which the author’s opinion about the contribution of Bm86 in immunoprotection was discussed taking into account our previous studies and also studies performed by other authors with other tick species.
- The English was also reviewed.
Round 2
Reviewer 1 Report
Comments and Suggestions for Authors
· Authors have implemented suggested modifications to the previous draft. I think that just a few more sentences should be added at certain places for better understanding and clarification, and minor corrections should be done (see below) to finalize the manuscript.
Line 40: based on
· Lines 67-69: “to avoid cross-recognition”
· Lines 70-71: maybe vaccine antigen instead of pharmaceutical active ingredient
· Lines 73-74: this is example of combined vaccine, can other antigen (here Bm86) be considered as an adjuvant…?; what does “considering its particulate nature” mean?
· Line 78: by immunization
· Fig. 2: (B) Engorgement parameters instead of phenotype
· Line 114: 1:1000 instead of 1000
· Table 1: maybe this title: Effects of immunization with pP0-Bm86 conjugate on challenge with I. ricinus ticks in rabbits
· Table 2: maybe this title: Effects of immunization with pP0-Bm86 conjugate on challenge with D. nitens ticks in horses
· Lines 199-217: Please rephrase a bit these sentences to be more clear
· Line 212: does “under conjugated conditions” mean “when conjugated to some other molecule”? If so, please change
· Line 232: please erase “which is one of the worst consequences of tick infestations”
· Line 244: two groups of three rabbits each, of both sexes…
· Paragraph 2.4 and lines 191-193: It should be stated that D. nitens is one-host tick, for better understanding; What is the effect of immunization with pP0-Bm86 conjugate on larvae placed on horses? (what does it mean “The effects on larva and nymph stages should be included in the effects on female ticks.”?)
· Regarding particular organs used: “The experiment could be done using the whole ticks, however, these organs were selected in order to improve sensibility and also because during initial feeding phase, it is suggested that P0 is a part of the biosynthetic machinery in the cells that play an important role in these tissues.” – you could add this in the text
· This should be added in the text also “the high dose that has been assayed in rabbits, dogs and cattle independently of their body masses assuming that these peptides are not potent as immunogens”
· This also should be added (in shorter form) for better understanding: “we have a source for Bm86 protein because our institution produces it in yeast, however, we don’t have P0 protein, only a synthetic peptide of P0. In previous experiments we had evidences that the peptide sticks poorly to the ELISA plate. For this reason, the protocol of this indirect ELISA was established using the peptide conjugated to a carrier protein different to that used for immunization. In this way, you only will detect specific antibodies against the P0 peptide. Obviously, specific antibodies against this non-related carrier protein should not be present in the animal sera immunized with other conjugate. Anyway, if some unspecific reactivity could be obtained, it would be blanked with the OD of pre-immune animal sera.”
· Line 348: serially twofold diluted
Comments on the Quality of English LanguageMinor English editing.
Author Response
Thanks again reviewer 1 to your suggestions for minor changes.
All suggested changes for English improvements were done and some sentences were rephrased to express ideas more clearly.
It was stated that D. nitens is one-host tick in the "materials and methods" section
Regarding particular organs used in RNAi experiment, a sentence was added explaining the important P0 role in these tissues.
The necessity to conduct dosage experiments was declared in the discussion section
A short explanation about the use of P0 conjugate used in the indirect ELISA was included in the "Materials and methods" section.